# MedIMeta: An easy-to-use meta-dataset for medical imaging applications

**Stefano Woerner**[1]                                        STEFANO.WOERNER@UNI-TUEBINGEN.DE
**Arthur Jaques**[1]
**Christian F. Baumgartner**[1,2]              CHRISTIAN.BAUMGARTNER@UNI-TUEBINGEN.DE
[1] *Cluster of Excellence "Machine Learning", University of Tübingen, Germany*
[2] *Faculty of Health Sciences and Medicine, University of Lucerne, Switzerland*

## Abstract

Scarcity of large, diverse, and well-annotated datasets remains a challenge in medical image analysis. Medical images vary in format, size, and other parameters and therefore require extensive preprocessing and standardisation for usage in machine learning. Addressing these challenges, we introduce the Medical Imaging Meta-Dataset (MedIMeta), a novel multi-domain, multi-task meta-dataset. MedIMeta contains 19 medical imaging datasets spanning 10 different domains and encompassing 54 distinct medical tasks, all of which are standardised to the same format and readily usable in PyTorch or other ML frameworks.

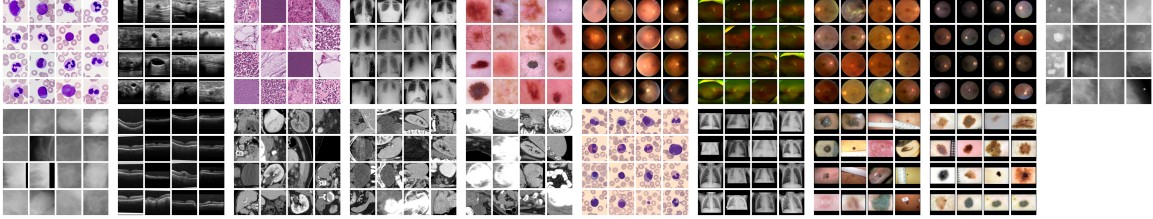

## 1. Introduction

Large, diverse and well-annotated datasets are pivotal for training robust and effective ML models and the advancement of the field of medical image analysis. However, the process collecting medical images and preparing them for ML applications is complex and fraught with challenges. This often presents a significant hurdle for researchers wanting to apply ML algorithms to medical imaging applications.

Addressing this issue, we introduce the Medical Imaging Meta-Dataset (MedIMeta), a novel multi-domain, multi-task meta-dataset designed to facilitate the development and standardised evaluation of ML models. In addition to the data, we release a user-friendly Python package to directly load images for use in PyTorch.

## 2. The MedIMeta Dataset

The MedIMeta dataset is comprised of 19 publicly available datasets containing a total of 54 tasks. All datasets have been previously published with an open license that allows redistribution or we obtained an explicit permission to do so. In contrast to the related MedicalMNIST dataset (Yang et al., 2023), all images were standardised to a size of $224 \times 224$ pixels. We also provide pre-defined train/val/test splits for all datasets. Most datasets include one main diagnostic task and several auxiliary tasks. The sub-datasets are:

**aml, AML Cytomorphology:** Single-cell leukocyte images from patients with and without acute myeloid leukemia (AML) from the *Munich AML Morphology Dataset* (Matek et al.). Contains a 15-way multi-class morphology classification task.

**bus, Breast Ultrasound:** Breast ultrasound images from the *Breast Ultrasound Images Dataset* (Al-Dhabyani et al.). Contains 2 tasks: tumor classification (3-way multi-class) and malignancy (binary).

**crc, Colorectal Cancer:** Image patches from hematoxylin & eosin (H&E) stained histological images of human colorectal cancer (CRC) and healthy tissue, from the *NCT-CRC-HE-100K* and *CRC-VAL-HE-7K* datasets (Kather et al.). Contains a 9-way multi-class tissue classification task.

**cxr, Chest X-ray Multi-disease:** Frontal-view X-ray chest images, from the *ChestX-ray14* dataset (Wang et al.). Contains a multi-label thorax disease classification task with 14 labels and a binary classification task of the patient sex.

**derm, Dermatoscopy:** Dermoscopic images of pigmented skin lesions from the *HAM10000* dataset (Tschandl et al., 2018). Contains a multi-class task with 7 disease classes.

**dr_regular, Diabetic Retinopathy (Regular Fundus):** Fundus photography images of patients with and without diabetic retinopathy from the *DeepDRiD* dataset (Liu et al., 2022). Contains 5 tasks: diabetic retinopathy grade (ordinal regression), sufficient image quality for gradability (binary classification), strength of artefact (ordinal regression), image clarity (ordinal regression), and field definition (ordinal regression).

**dr_uwf, Diabetic Retinopathy (Ultra-widefield Fundus):** Ultra-widefield fundus images of patients with and without diabetic retinopathy, from the *DeepDRiD* dataset (Liu et al., 2022). Contains a DR grading task (ordinal regression) with 5 labels.

**fundus, Fundus Multi-disease:** Retinal fundus images with 45 conditions, from the *Retinal Fundus Multi-disease Image Dataset* (Pachade et al., 2021). Contains 2 tasks: disease presence (binary) and disease type (multi-label classification with 45 labels).

**glaucoma, Glaucoma-specific fundus images:** Fundus photography images with patients with and without glaucoma from the *Chákṣu* dataset (Kumar et al., 2023). Contains a glaucoma suspect binary classification task.

**mammo_calc, Mammography (Calcifications):** Cropped calcification regions obtained from mammography images from CBIS-DDSM (Lee et al.). Contains 3 tasks: malignancy (binary), calcification type (multi-label with 14 labels), and calcification distribution (multi-label with 5 labels).

**mammo_mass, Mammography (Masses):** Cropped mass regions obtained from mammography images from CBIS-DDSM (Lee et al.). Contains 3 tasks: malignancy (binary), mass shape (multi-label with 8 labels), and mass margins (multi-label with 5 labels).

`oct`, **OCT:** OCT images from (Kermany et al.). Contains a multi-class disease classification task with 4 labels and a binary task for predicting urgent referral.

`organs_axial`, **Axial Organ Slices:** Cropped axial image slices of 11 different organs, extracted from the LiTS dataset (Bilic et al., 2023) and the organ bounding boxes from (Xu et al., 2019). Contains a multi-class organ classification task with 11 labels.

`organs_coronal`, **Coronal Organ Slices:** Cropped coronal image slices of 11 different organs, containing the coronal projections of the same subjects as `organs_axial`.

`organs_sagittal`, **Sagittal Organ Slices:** Cropped sagittal image slices of 11 different organs, containing the sagittal projections of the same subjects as `organs_axial`.

`pbc`, **Peripheral Blood Cells:** Microscopic peripheral blood cell images of normal cells and cells with hematologic or oncologic disease from (Acevedo et al.). Contains a multi-class blood cell classification task with 8 labels.

`pneumonia`, **Pediatric Pneumonia:** Pediatric chest X-ray images labeled for pneumonia, from (Kermany et al.). Contains two tasks: pneumonia presence (binary) and disease class (multi-class between normal, bacterial pneumonia and viral pneumonia).

`skinl_photo`, **Skin Lesion Evaluation (Clinical Photography):** Clinical colour photography images of skin lesions from (Kawahara et al., 2019). Contains an overall diagnostic multi-class task, as well as classification tasks for each diagnostic criterion.

`skinl_derm`, **Skin Lesion Evaluation (Dermoscopy):** Dermoscopic colour images of skin lesions. This dataset contains the same subjects, labels and tasks as `skinl_photo`.

## 3. Validation

In order to validate our proposed dataset, we trained ResNet-18 and ResNet-50 models (He et al., 2016) on the primary task for each dataset. All networks were initialised with pre-trained weights from ImageNet (Russakovsky et al., 2015). The results are shown in Table 1.

Table 1: AUROC (%) on the test set for the fully supervised baselines.

| | aml | bus | crc | cxr | derm | dr_regular | dr_uwf | fundus | glaucoma | mammo_calc |
|---|---|---|---|---|---|---|---|---|---|---|
| ResNet 18 | **99.1** | 96.0 | **99.5** | 78.4 | 94.0 | 84.9 | 58.1 | 97.1 | 74.5 | **76.8** |
| ResNet 50 | 98.4 | **96.6** | **99.5** | **80.5** | **96.2** | **86.5** | **58.3** | **97.2** | **86.6** | 75.0 |

| | mammo_mass | oct | organs_axial | organs_coronal | organs_sagittal | pbc | pneumonia | skinl_photo | skinl_derm |
|---|---|---|---|---|---|---|---|---|---|
| ResNet 18 | **73.0** | 99.7 | 97.8 | 97.1 | 96.7 | 99.9 | 98.7 | 75.0 | 82.9 |
| ResNet 50 | 72.3 | **99.8** | **98.6** | **97.9** | **96.8** | **100.0** | **99.3** | **78.0** | **90.1** |

## 4. Usage

The MedIMeta dataset can be downloaded from Zenodo (Woerner et al.). Our data loaders and examples repository[1] provides simple code for loading all tasks as PyTorch datasets. Our contributions will substantially simplify the development and evaluation of ML algorithms for medical image analysis, and will allow to easily transferring algorithms to different tasks.

**Acknowledgements.** Funded by DFG grant EXC 2064/1 – Project number 390727645.

---

1. https://github.com/StefanoWoerner/mimeta-pytorch

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
