# OpenReview forum: "MedIMeta: An easy-to-use meta-dataset for medical imaging applications"
_MIDL.io/2024/Short_Papers — MIDL 2024 Short Papers_

### Official Review · Reviewer_Sg7J · 2024-04-18

**Confidence:** 5
**Final Rating:** 4

**Review:**

The paper presents a collection of datasets with different domains and different tasks, which have been standardized in the same way for benchmarking machine learning algorithms.

Strengths:

The paper presents a collection of multi-task datasets from different domains, which have been standardized in the same way for benchmarking machine learning algorithms.

Strengths:
-	Important for the community to focus more on datasets
-	Code for loading the datasets
-	Paper clear to read


Weaknesses:
-	Providing predefined splits for the tests reinforces poor evaluation practices, researchers will be overfitting on these splits – some caution should be adviced here
-	Not sure about the motivation of seeing the coronal/saggital/axial slices as different datasets, this seems to break some assumptions you would want in benchmarking that datasets are independent
-	Details about the datasets are missing (patient population, data sampling/generation), this would not fit in the short paper but should be provided in the READMEs of the repositories.

Overall I think this is definitely worth presenting/discussing at the conference, but there are some caveats the authors should be aware of to avoid perpetuating bad evaluation practices.

---

### Decision · Program_Chairs · 2024-04-26

Accept